# Catalytic Deacidification of Vacuum Gas Oil by ZnO/Al$_2$O$_3$ and Its Modification with Fe$_2$O$_3$

**Bai He [1,2], Xue Fu [2], Xin Lian [2], Songshan Jiang [2], Peng Xu [2], Xinting Deng [3], Changxuan He [2] and Changguo Chen [1,*]**

[1]  School of Chemistry and Chemical Engineering, Chongqing University, Chongqing 401331, China; hbai2004@126.com
[2]  College of Chemistry and Chemical Engineering, Chongqing University of Science and Technology, Chongqing 401331, China; fuxue1981@163.com (X.F.); daisylian0121@163.com (X.L.); cessjiang@cqust.edu.cn (S.J.); xpthomas@163.com (P.X.); hechangxuan126@126.com (C.H.)
[3]  ChongQing NatruGas Purification Plant General, PetroChina Southwest Oil & Gasfield Company, Chongqing 400021, China; dengxinting@petrochina.com.cn
*  Correspondence: cgchen@cqu.edu.cn

**Abstract:** High acidic vacuum gas oil (VGO) decreases product quality and causes corrosions. To overcome these problems, catalytic esterification was utilized for the deacidification of VGO. A series of Al$_2$O$_3$-supported ZnO catalysts, undoped and doped with Fe$_2$O$_3$, were prepared and characterized. The results showed that both the ZnO/Al$_2$O$_3$ and the ZnO/Al$_2$O$_3$ doped with Fe$_2$O$_3$ had good deacidification effects with glycol for the 4th VGO. The deacidification rate reached 95.1% and 97.6%, respectively, under mild conditions (catalysts 2.5 wt%, glycol dosage 4.0 wt%, 250 °C and 1 h). The naphthenic acids were transferred into ester, which was proved by the Fourier Transform Infrared (FT-IR) and $^1$H nuclear magnetic resonance (NMR).Reusability of the catalyst for the esterification reaction was also studied. It was found that the deacidification rate was still over 90% after six reruns.

**Keywords:** 4th vacuum gas oil; catalytic esterification; deacidification; glycol

## 1. Introduction

Worldwide, high total acid number (TAN) crude oil production has increased in recent years, leading to high TAN petroleum fractions, especially vacuum gas oil (VGO). The total acid numbers (TANs) of crude oil and petroleum fractions are mainly supplied by naphthenic acids (NAs), and this refers to all carboxylic acids present in the oil which have the general formula RCOOH, where R represents cyclopentane and cyclohexane derivatives. The NAs may cause serious equipment failures, lead to high maintenance costs, reduce product quality, and pose environmental disposal issues [1]. For these reasons, the removal of intrinsic NAs from crude oil and petroleum fractions is of paramount necessity. Various techniques have been put forward to solve this difficult problem, such as a treatment using ionic liquids, catalytic esterification, catalytic decarboxylation, neutralization or caustic washing, thermal decomposition, physical adsorption, and solvent extraction [2]. However, some of these approaches have drawbacks. For example, the ionic liquid treatment, catalytic hydrogenation, physical adsorption, and thermal decomposition are usually costly, because of expensive chemicals or high energy consumption. Catalytic decarboxylation needs a high temperature for operation and metal oxide catalysts could be polluted by acid–base neutralization reactions with the NAs. Neutralization or caustic washing might cause emulsion problems and product loss. Solvent extraction requires a large amount of solvent, which is complex and needs lots of energy for solvent recovery [3–6]. So, it is important to adopt a simple, high efficiency, environmentally-friendly, and economical deacidification technology to remove NAs from the crude oil and fractions.

In the present work, esterification is an interesting and promising way to deal with removal of the NAs into naphthenic acid esters without loss of oil, which overcomes the disadvantages of the traditional deacidification methods for crude oil and petroleum fractions [7], and makes the process suitable for existing refineries. The non-catalytic deacidification method was adapted to process the high-acid crude via esterification with methanol, in which the reaction occurred under a high pressure of up to 6.4 mPa and with the needed amount of methanol [8]. Considering most of the NAs from crude oils will transfer into VGO in refiners, the deacidification of VGO plays a significant role in the process. Some researchers have processed petroleum fractions by catalytic esterification and achieved a deacidification efficiency of up to 95.6%. However, the reaction time took 6 hours [1,9,10]. Among the catalysts tested for catalytic esterification, SnO and ZnO were the most popular, with oxide loaded on different supporters such as $SnO/Al_2O_3$, $ZnO/SiO_2$ [11], ZnO/zeolite [12,13], and $ZnO/SiO_2$ [11,14,15]. In particular, the ZnO was found to be a relatively economical, non-toxic, and environmentally friendly material. The esterification reactions using zinc oxide supported on alumina was only used for phytosteryl esters from the esterification production of phytosterol with fatty acids [16]. However, this catalyst was not used for the esterification deacidification of naphthenic acids that are present in hydrocarbon fractions and acidic crude oil. In a recent study [17], we applied $ZnO/Al_2O_3$ as a catalyst for 4th VGO esterification deacidification under moderate reaction conditions, and the deacidification rate was over 92% while other main physicochemical properties varied little. Nevertheless, the effects of preparation parameters on the deacidification rate have not been investigated sufficiently. Furthermore, the modification of a $ZnO/Al_2O_3$ catalyst with small amounts of $Fe_2O_3$ has not been reported [18]. In this regard, the aim of this study is to prepare $ZnO/Al_2O_3$ solid acid catalysts under different conditions and modify the catalysts by $Fe_2O_3$ to obtain higher deacidification rates of VGO. The general reaction scheme of the esterification process is given as the Scheme 1 below [7].

$$R-[\pentagon]_n(CH_2)_mCOOH \ + \ R'OH \ \xrightarrow[250℃]{cat.} \ R-[\pentagon]_n(CH_2)_mCOOR' \ + \ H_2O$$

**Scheme 1.** Proposed esterification reaction of the naphthenic acid and glycol.

## 2. Results and Discussion

### 2.1. Characterization of Catalyst

#### 2.1.1. XRD and Phases

Figure 1 shows X-ray diffraction (XRD) patterns of the synthesized catalyst. When the calcination time was extended to 45 min and the calcination temperature was 400 °C, typical diffraction peaks of ZnO were observed, as shown in Figure 1a, indicating the formation of a ZnO crystal with a hexagonal structure [19]. The crystallinity increased with the calcination time because it needed enough time for the decomposition of $Zn(NO_3)_2$ into ZnO, as well as the growth of the crystal. However, long-time calcinations had no benefit for the ZnO crystal at high temperatures, and it led to a drop of diffraction peaks in all directions, due to the probable fusion into the carrier. Further investigation showed that the main structure of $Al_2O_3$ remained unchanged until the calcination time was 3 h. The existence of an $Al_2O_3$ crystal was demonstrated by XRD diffraction peaks at 2θ of 14.4°, 28.3°, 38.5°, and 48.8°. The effect of the calcination temperature on the crystallinity is depicted in Figure 1b. It was observed that 400 °C was the best calcination temperature for the crystal of the ZnO, and the diffraction peaks of ZnO was most clear at 2θ = 31.9°, 34.7°, 36.4°, 47.7°, 56.8°, and 63.10°, which corresponded to the lattice planes (100), (002), (101), (102), (110), and (103), respectively [4]. The decomposition temperature of the $Zn(NO_3)_2$ was around 350 °C, and an appropriately higher temperature was a benefit for the complete decomposition of $Zn(NO_3)_2$. The effect of ZnO loading on the crystallinity is shown in Figure 1c. It indicated that 5% to even 10% loading of ZnO was not enough for ZnO crystal formation, and a higher loading of ZnO was beneficial for the ZnO crystal with sharper diffraction intensity. When the ZnO



loading was up to 25% or more, the diffraction peaks of $Al_2O_3$ dropped sharply. The XRD pattern showed almost only typical diffraction peaks of ZnO, indicating that the surface of the carrier was covered by plenty of ZnO crystal. Moreover, two mixed metal oxides were found on the XRD patterns, especially when the loading of $Fe_2O_3$ was increased over to 3.0%, as seen in Figure 1d. The peak of $Fe_2O_3$ at $2\theta = 33.1°$ ostensibly appeared [20], implying that an approximately 3.0% amount of $Fe_2O_3$ was necessary for the $Fe_2O_3$ crystal's growth.

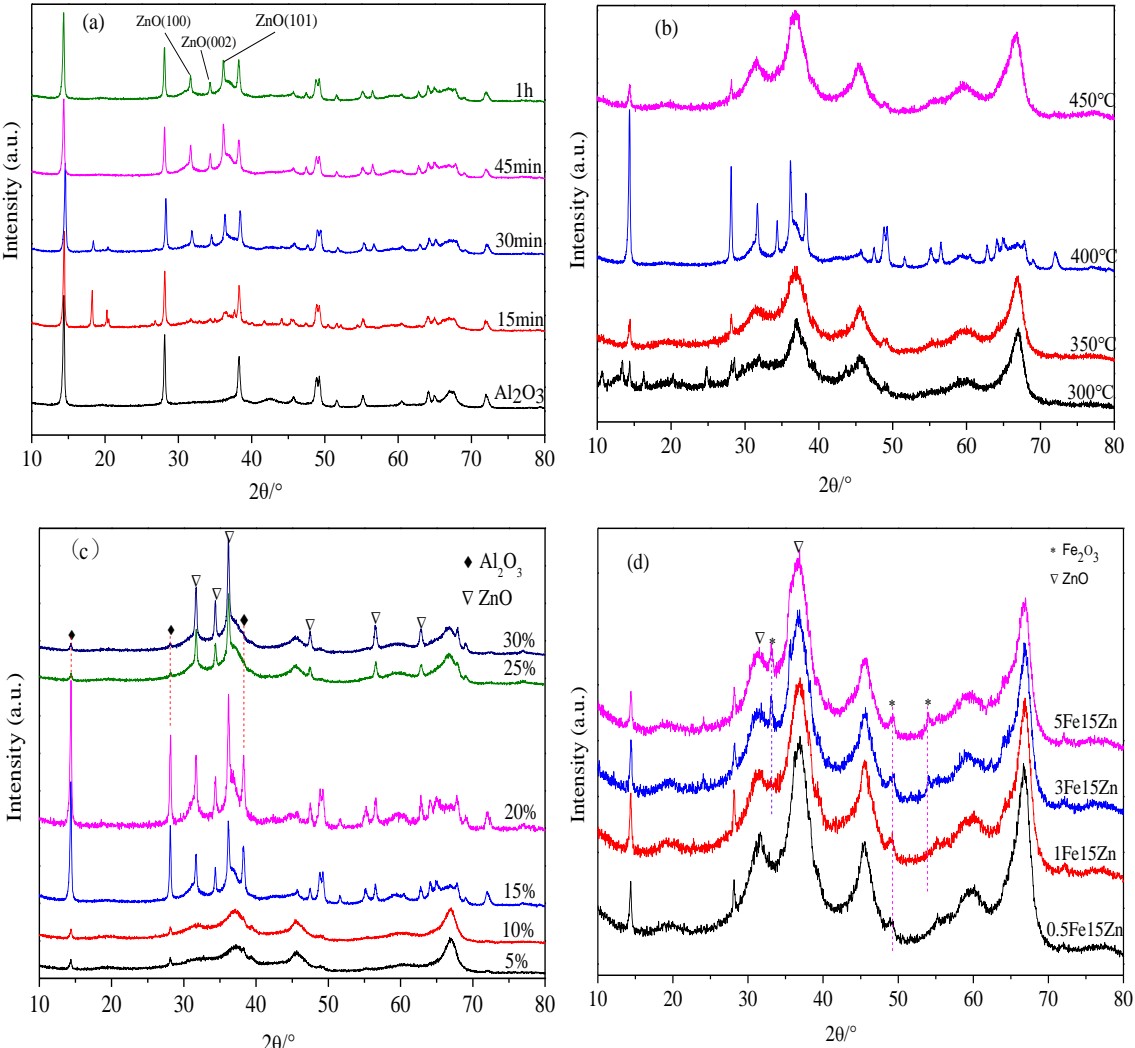

**Figure 1.** XRD patterns of catalysts with different calcination time, calcination temperature, ZnO loading, and $Fe_2O_3$-doped amount. (**a**) Different calcination time; (**b**) different calcinations temperature; (**c**) different ZnO loading; (**d**) different $Fe_2O_3$-doped amount.

2.1.2. Brunauer Emmett Teller (BET) Surface Area and Pore Volume

The $N_2$ adsorption isotherms results are shown in Figure 2 and the BET surface areas of the synthesized catalysts particles are shown in Table 1. The results indicate the effects of different metal oxide loading on the BET surface areas and pore volumes. The blank $Al_2O_3$ had the maximum BET surface areas and pore volumes. When the ZnO loading amount was 15%, both the BET surface areas and pore volumes dropped off, because the ZnO particles covered part of the small pores and led to a reduction of pore surface area and pore volume. Zinc oxide catalysts promoted with various $Fe_2O_3$ amounts showed that the higher the $Fe_2O_3$ loads, the less BET surface area and pore volume. Greater amounts of $Fe_2O_3$ were a benefit for forming bigger crystals, which had less BET surface area and pore volume than the small particles. Figure 2 shows the $N_2$ adsorption isotherms' results of the carrier

blank $Al_2O_3$ and the various catalysts. The characteristic features of the physisorption isotherms in Figure 2 belong to Type IV with hysteresis loops. This is associated with limiting uptake over a range of high $P/P_0$, which means that the pore size distribution of the catalysts used in this work is mainly mesopore where capillary condensation has taken place. The hysteresis loops in Figure 2 were Type H2, which was attributed to a difference in mechanism between condensation and evaporation processes occurring in pores with narrow necks and wide bodies, indicating that the pores of the catalysts were proven to be "ink bottle" pores [21]. In addition, the pore size distribution also confirmed the mesopore diameter of the catalysts, as shown in Figure 2.

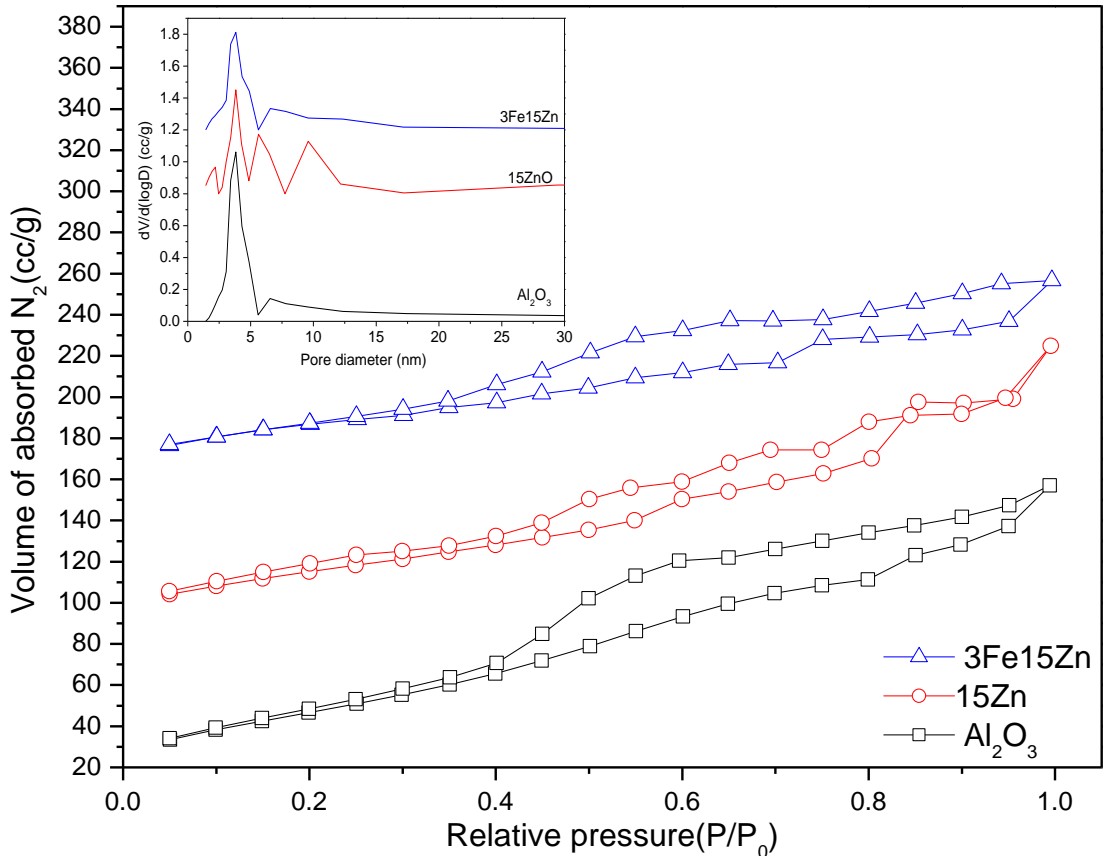

**Figure 2.** The $N_2$ adsorption-desorption isotherms and pore size distribution (inset) of different catalysts.

**Table 1.** Pore structure parameters of the different catalysts.

| Samples | $Al_2O_3$ | 15Zn | 0.5Fe15Zn | 1Fe15Zn | 3Fe15Zn |
|---|---|---|---|---|---|
| BET areas (m²/g) | 174 | 132 | 132 | 131 | 129 |
| Pore Volume (cm³/g) | 0.24 | 0.23 | 0.23 | 0.17 | 0.16 |

### 2.1.3. SEM Analysis

The effect of calcination time on the particle images is shown in Figure 3a–c. Results showed that 45 min was the optimal calcination time for the ZnO particle dispersity when the calcination temperature was 400 °C and with a the ZnO loading amount of 15%, and the average grain size of ZnO was about 40 nm. In fact, although sufficient time is good for the crystal growth, longer calcination times will lead to the fusion of ZnO with the carrier aluminum oxide as shown in Figure 3c. It can be seen from the Figure 3d that if the calcination temperature was up to 450 °C, only a small part of the ZnO particles could be seen, because of the fusion into the carrier. In Figure 3e–f, there were no apparent compact ZnO particles, because the amount of zinc oxide was not enough when the

ZnO loading amount was 5%. When the ZnO loading amount was increased to 30%, a large block of small ZnO particles formed and almost completely coated the aluminum oxide carrier. These phenomena were consistent with the XRD results provided in Figure 1c. The surface morphology of $Fe_2O_3$-$ZnO/Al_2O_3$ is shown in Figure 3g–i. The particle surface seemed to be loose and porous with the $Fe_2O_3$-doped amount of 0.5%. When the $Fe_2O_3$ amount increased to 3% or even 5%, the $Fe_2O_3$-doped $ZnO/Al_2O_3$ catalyst presented similar images, and more $Fe_2O_3$ contributed to relatively bigger and regular composite particles.

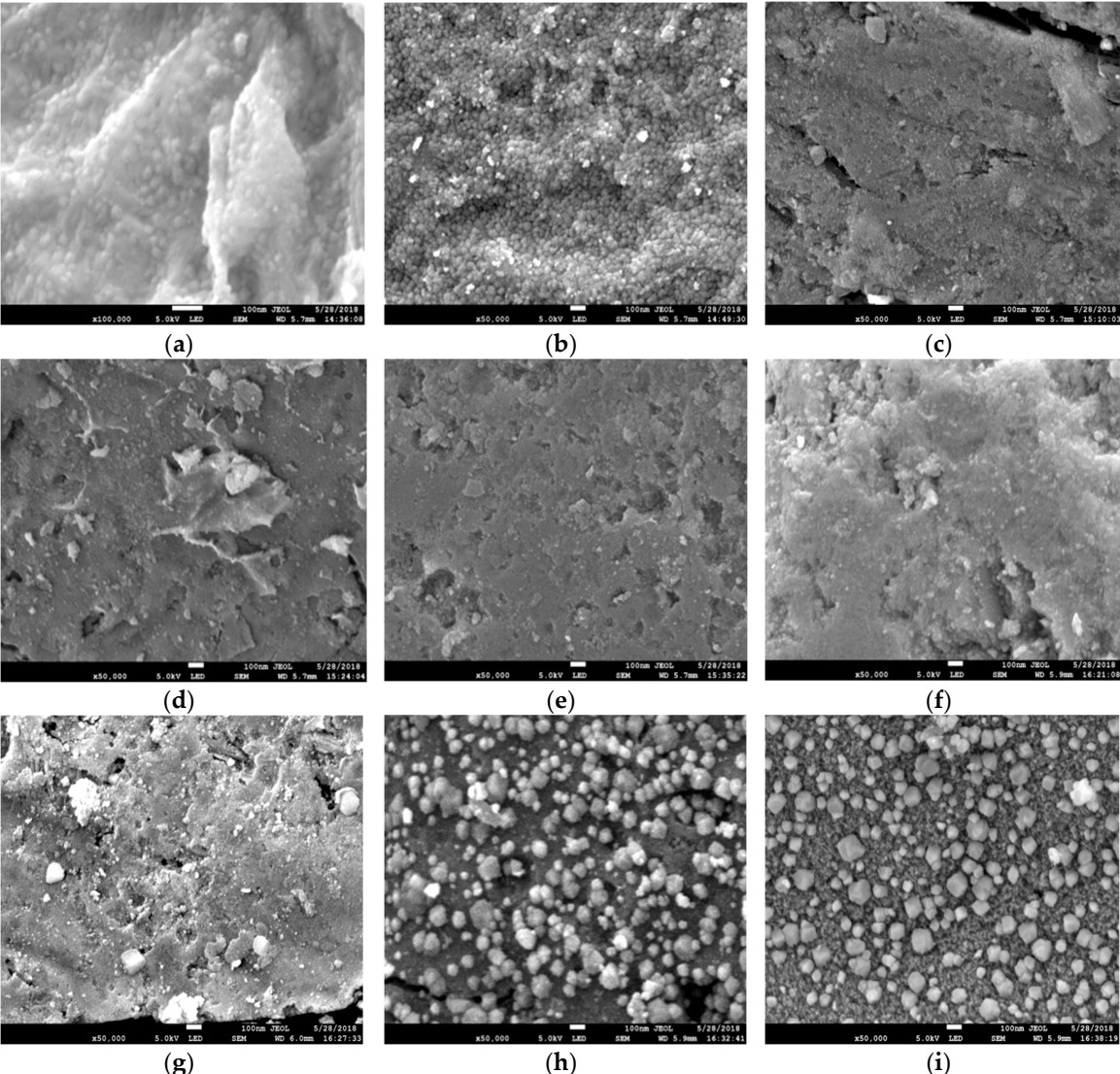

**Figure 3.** SEM images of catalysts prepared under different conditions. (**a**–**c**) 15Zn, calcination temperature 400 °C, with calcination times of 30, 45, and 300 min, respectively; (**d**)15Zn, calcination temperature 450 °C, calcination time 45 min; (**e**–**f**) calcination temperature 400 °C, calcination time 45 min, with 5Zn and 30Zn, respectively; (**g**–**i**) calcination temperature 400 °C, calcination time 45 min, with 0.5Fe15Zn,3Fe15Zn and 5Fe15Zn, respectively.

2.1.4. Temperature-Programmed Desorption (TPD)

The acidities of the catalysts 15Zn and 0.5Fe15Zn were determined by $NH_3$-TPD, as depicted in Figure 4. Table 2 lists the $NH_3$-TPD amounts. Obviously, three $NH_3$ desorption curves were detected in the ranges of 100–600 °C with two desorption peaks, indicating the weak and strong acid sites on the surface of the catalyst, respectively. Moreover, the number of strong acid sites was larger than that

of weak ones for 15Zn, while the weak acid sites were slightly more numerous than those of the strong for the 0.5Fe15Zn. There are more weak acid sites on 0.5Fe15Zn compared to 15Zn, although the higher total acid sites of the latter had been observed.

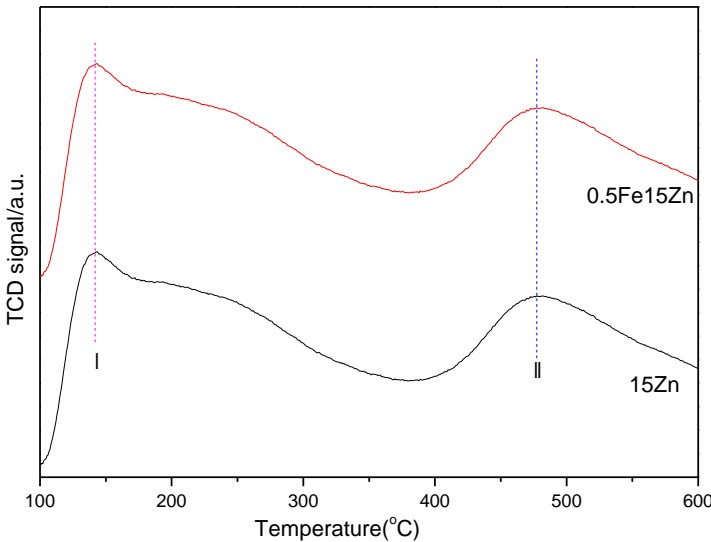

**Figure 4.** NH$_3$-TPD profiles of the catalysts 15Zn and 0.5Fe15Zn.

**Table 2.** NH$_3$-TPD results of the catalysts.

| Catalysts | NH$_3$ Desorption: Peak I/(mmol·g$^{-1}$) | NH$_3$ Desorption: Peak II/(mmol g$^{-1}$) | Total NH$_3$ Desorption/(mmol g$^{-1}$) |
|---|---|---|---|
| 15Zn | 0.27 | 0.48 | 0.75 |
| 0.5Fe15Zn | 0.29 | 0.25 | 0.54 |

## *2.2. Influencing Factors of the Esterification Reaction*

### 2.2.1. Effect of Calcination Time

The effect of the calcination time on the reduction of the TAN was investigated in order to determine the optimum catalyst preparation conditions at the calcination temperature of 400 °C and loading amount of 15% ZnO. The results are summarized in Figure 5. When calcination time was extended from 15 min to 45 min, the deacidification rate increased from 62.3% to 95.1% because enough time was needed for the decomposition of the Zn(NO$_3$)$_2$ into the active component ZnO. Then the deacidification rate began to decline with longer calcination times, because long-time calcinations at high temperatures would not only lead to fusion of ZnO, which could reduce the effective contact between the reactant and active constituent ZnO, but also decrease the surface acid amount of the catalysts, which was not benefit for the esterification reaction [17]. Thus, the optimal calcination time was 45 min.

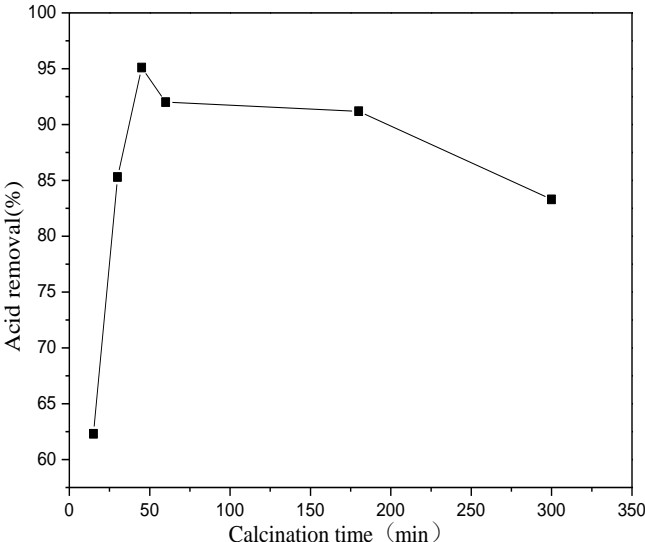

**Figure 5.** Effect of catalyst calcination time on the esterification reaction.

### 2.2.2. Effect of Calcination Temperature

The calcination temperature is an important parameter for the morphology, structure, and properties of the catalyst. The effect of calcination temperature on reduction of the TAN was studied at the calcination time of 45 min with a 15% loading amount of ZnO. The results are shown in Figure 6. The deacidification rate first increased from 82.4% to 95.1%, then reduced to 82.6% as the calcination temperature varied from 300 °C to 450 °C. This indicates that a proper calcination temperature was required for effective deacidification of the VGO, because low temperature was unfavorable for the decomposition of $Zn(NO_3)_2$ and corresponded to crystal formation of ZnO. Furthermore, the catalyst surface area decreased because of the long-time high temperature calcinations [22], which was not beneficial for the esterification reaction and deacidification rate. Therefore, a proper calcination temperature was determined to be 400 °C.

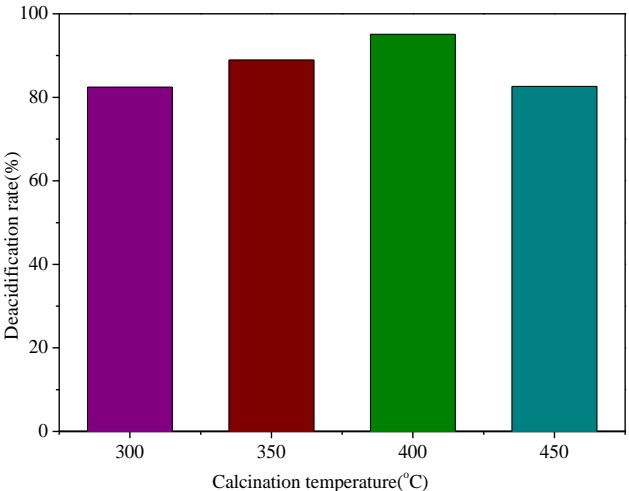

**Figure 6.** Effect of different catalyst calcination temperature on the esterification reaction.

### 2.2.3. Effect of ZnO Loading

As the active species, a series of catalysts with different ZnO loading amounts were applied to the esterification of the VGO and glycol, as seen in in Figure 7. The deacidification rate increased from 82.1% to 90.4%, and then reached the maximum of 95.1% with the ZnO loading amount increased to

15%. This is probably because more active constituent was beneficial for the esterification reaction. Furthermore, a too-large amount of ZnO on the carrier decreased the deacidification rate. Due to the gathering phenomena caused by the excessive ZnO, the full cover of the carrier and the surface area and the pores of the catalyst were reduced. Therefore, the proper ZnO loading amount should be 15%.

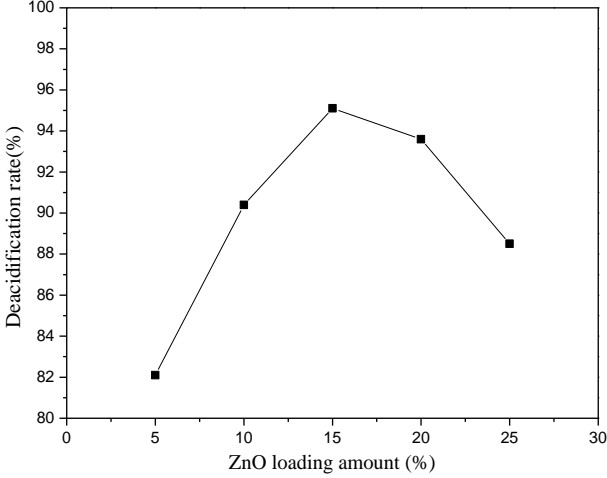

**Figure 7.** Effect of different ZnO loading amount on the esterification reaction.

### 2.2.4. Effect of $Fe_2O_3$ Doped Amount

In order to further improve the catalytic performance of the zinc oxide catalyst, doping with $Fe_2O_3$ was tested, and the deacidification rates of the modified catalysts are shown in Figure 8. The deacidification rate was 97.6% by using the composite catalyst with a $Fe_2O_3$-doped amount of 0.5%. $Fe_2O_3$ addition could help to improve the esterification rate, because there are more surface weak acid sites on $Fe_2O_3$ than that of ZnO (in Table 2), which is a benefit for the esterification reaction [23–25], so the $Fe_2O_3$-doped catalyst resulted in a higher deacidification rate. Then, the deacidification rate changed little in spite of more $Fe_2O_3$, and the maximum rate was 98.4%. More $Fe_2O_3$ did not help to increase the deacidification rate greatly, which was probably because more weak acid sites contributed to a higher esterification effect. But more $Fe_2O_3$ decreased the total surface areas and pore volumes of the catalysts (shown in Figures 2 and 3), so it was not good for the esterification reaction. Thus, doping with a little ferric oxide was reasonable for the overall deacidification effect.

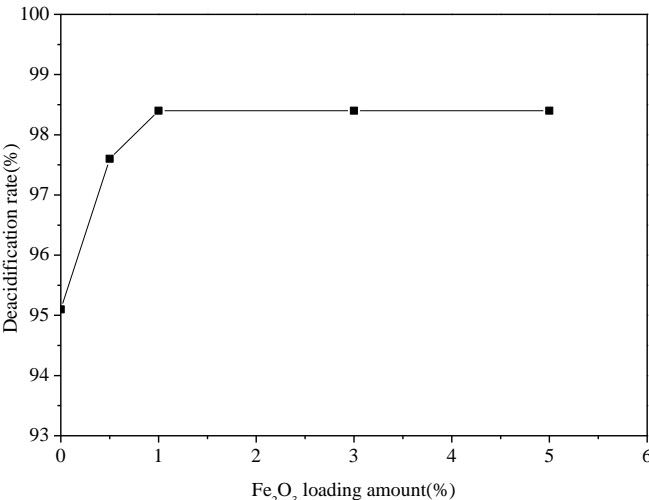

**Figure 8.** Effect of different $Fe_2O_3$-doped amounts on the esterification reaction.

### 2.2.5. Effect of Impregnation Time

The impregnation time affects the efficiency of the catalyst preparation, so different impregnation times were observed (as shown in Figure 9). The deacidification rate was 89.2% at the corresponding impregnation time of 3 h, but when the impregnation time was extended to 12 h, the deacidification rate increased to 97.6% and did not increase any more. Therefore, to make sure of the final high deacidification rate and efficiency of the catalysts' preparation, an impregnation time of 12 h was found to be appropriate.

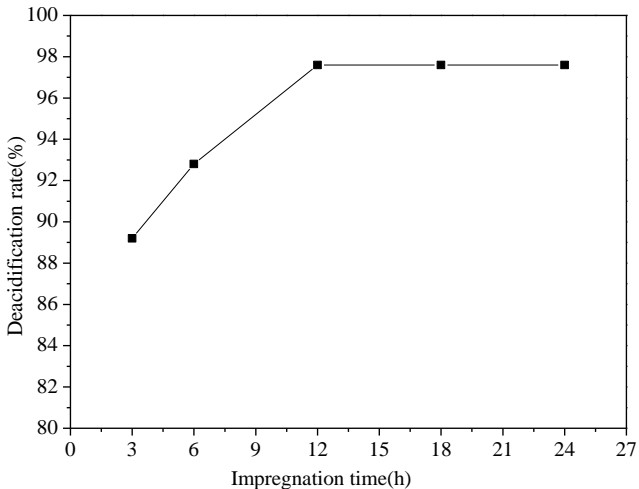

**Figure 9.** Effect of different impregnation times on the esterification reaction.

### 2.3. FT-IR and $^1$H NMR Spectra before and after the Esterification

To analyze the difference before and after the esterification reaction of the raw VGO, FT-IR spectroscopy was used to identify the products catalyzed by $ZnO/Al_2O_3$(15Zn) and $Fe_2O_3$–$ZnO/Al_2O_3$(0.5Fe15Zn), to be compared with the raw VGO. The results are presented in Figure 10. As for the raw VGO, the peak at approximately 1705 cm$^{-1}$ is the characteristic absorption of carboxyl group (–COOH) [26]. Compared with the raw VGO, the absorption observed at 1740 cm$^{-1}$ is associated with ester function (COOC) [27] after the esterification reaction. The carboxyl functional group at 1705 cm$^{-1}$ almost disappeared, which indicated that the NAs were converted into esters.

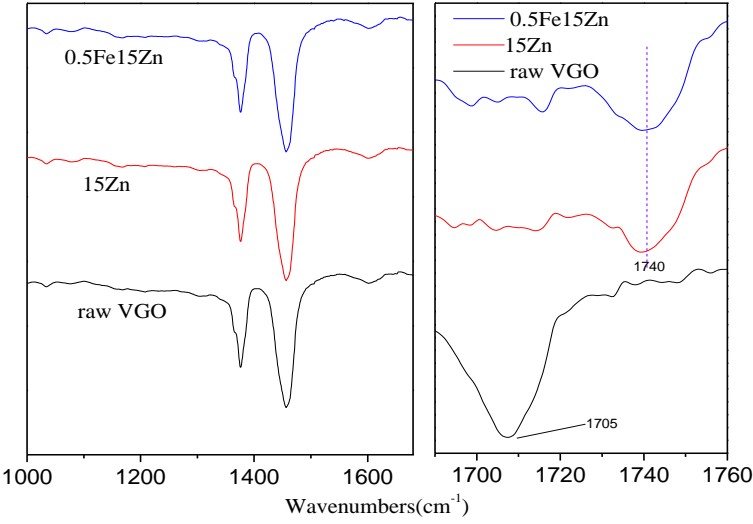

**Figure 10.** FT-IR spectra of raw and catalytic esterified vacuum gas oil (VGO).

The FT-IR spectral data was certificated by the $^1$H NMR spectra of the esterified VGO catalyzed by 0.5Fe15Zn, as shown in Figure 11. The two peaks appear at 3.84 ppm and 4.23 ppm with the ratio of the integral curve area 1:1, which belongs to the ethylene of the esterified glycol with the NAs. Those two signals could be assigned to the protons of $-CO_2-CH_2-CH_2-O_2C-$. The weak signal at 4.29 ppm could be assigned to the ethylene protons of $-CO_2-CH_2-CH_2-OH$. However, the other part of the signal of ethylene protons of $-CO_2-CH_2-CH_2-OH$ was covered by the peak at 3.84 ppm. The vanishing of the carboxyl signal with the appearance of those signals from 3.84 ppm to 4.29 ppm suggested that the esterification of the carboxyl group was accomplished.

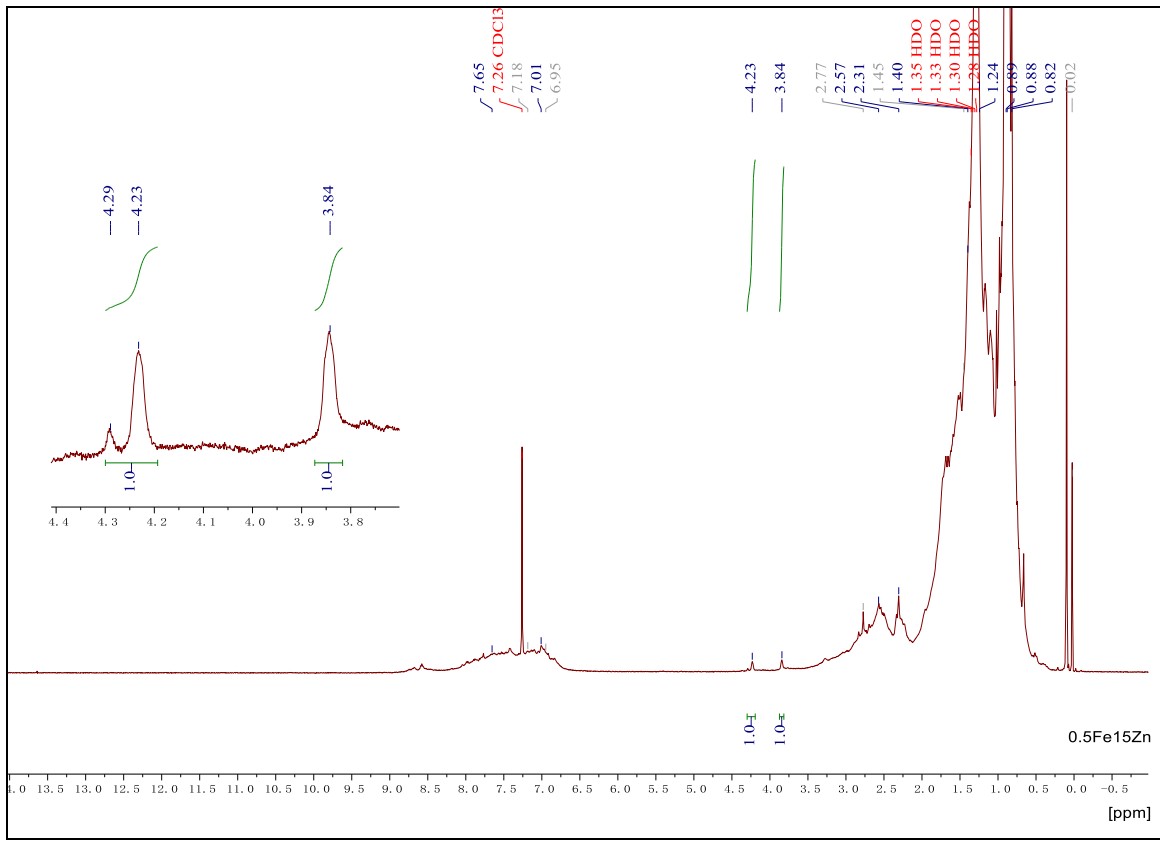

**Figure 11.** $^1$H NMR spectra of a mixture of catalytic esterified VGO.

## 2.4. Reusability of the Catalyst

The recycling performance of the selected catalysts was investigated by performing several repeat reactions under the optimum conditions. The results presented in Figure 12 showed a slight decline in catalytic performance, and the deacidification was 90.4% after six successive reruns. Compared with traditional catalysts, the possibility of efficiently recycling $Fe_2O_3-ZnO/Al_2O_3$ was of interest from environmental and economic perspectives.

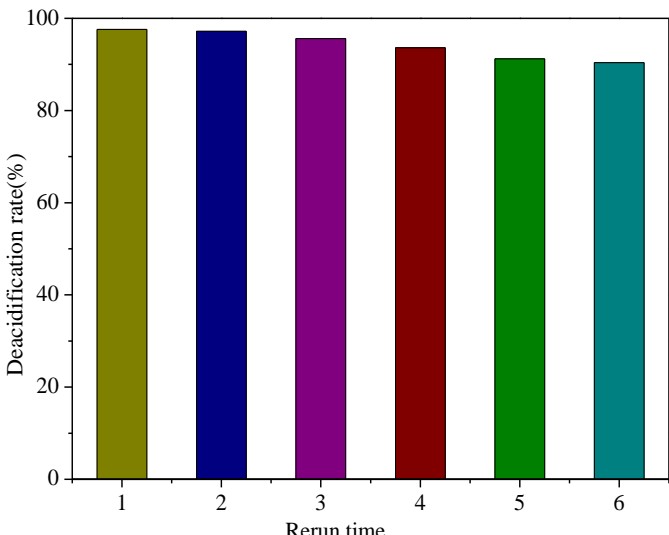

**Figure 12.** Reusability of the catalyst for the esterification.

## 3. Experimental Section

### 3.1. Materials

The 4th VGO (TAN 2.51 mgKOH/g) was obtained from COCC Asphalt (Sichuan) Ltd., Sichuan, China. The aluminum oxide was purchased from China Medicine (Group) Shanghai Chemical Reagent Co., Ltd., Shanghai, China. Zinc nitrate, ferric nitrate, and glycol was obtained from Chengdu Kelong Chemical Reagent Factory, Chengdu, China.

### 3.2. Catalyst Preparation

$Al_2O_3$ were impregnated with an appropriate amount of aqueous $Zn(NO_3)_2$ solution to obtain various ZnO loading amounts on the $Al_2O_3$. The mixture was stirred at room temperature for 10 min, and then dried at 120 °C after laying up 6 h. The catalyst ($ZnO/Al_2O_3$) was calcined in air at different temperatures and times (by putting the sample into the muffle furnace for the set time as the temperature rose up to the preset value). The $Fe_2O_3$-promoted zinc oxide catalysts were prepared by a co-precipitation method using aqueous solutions of $Zn(NO_3)_2 \cdot 6H_2O$ and $Fe(NO_3)_3 \cdot 9H_2O$. After the aforementioned drying and calcinations, the modified catalyst of $Fe_2O_3$-$ZnO/Al_2O_3$ could be obtained. The catalyst $ZnO/Al_2O_3$ with ZnO loading of 15% (weight percentage based on the carrier, the same as below) was denoted as 15Zn. The $Fe_2O_3$-$ZnO/Al_2O_3$ with $Fe_2O_3$ loading of 0.5% and ZnO loading of 15% was denoted as 0.5Fe15Zn and the other catalysts were similarly denoted.

### 3.3. Characterization

XRD patterns were recorded with a Shimadzu XRD-7000 (Kyoto, Japan) using Cu K$\alpha$ radiation ($\lambda$ = 1.5418 Å, 40 kV, 30 mA), and a preset time of 2 s. SEM images were conducted on a JSM-7800F microscope (JEOL Ltd., Kyoto, Japan) operated at 20 kV landing energy. The BET surface areas were calculated from nitrogen adsorption data at 77.35 K obtained using Quanta chrome Autosorb iQ (Quantachrome Instruments, Boynton Beach, FL, USA). The outgas time was 6 h and the final out gas temperature was 200 °C. The acidity of the catalysts were studied via ammonia temperature-programmed desorption ($NH_3$-TPD) using AutoChem II 2920 (Micromeritics, Norcross, GA, USA) and the carrier gas was He. The acid number of the raw VGO or after the esterification reaction was determined according to the American Society of Testing and Materials (ASTM) D664 international standard method. FT-IR spectra were acquired using a Nicolet iS50 FT-IR Spectrometer (Thermo Scientific, Waltham, MA, USA) and the measurements were performed in the transmission mode from 4000 and 400 cm$^{-1}$ with an

accumulation of 32 scans at a nominal resolution of 4 $cm^{-1}$. $^1H$ NMR spectra were obtained at room temperature on a Bruker (Billerica, MA, USA) AVANCE 500 MHz and the solvent was $CDCl_3$.

*3.4. Experimental Procedures*

The reactor was a 250 mL three-necked round-bottom flask, connected with a water-cooled condenser. The reactor was charged with 40 g VGO, glycol 4 wt%, and catalysts 2.5 wt%. The charged reactor was then placed in a thermostatic bath and heated to the 250 °C for 1 h. After the reaction, the produced water and the residual glycol could evaporate away at a temperature of 250 °C. When the flask was cooled down to the room temperature in the air, there was no obvious water observed. Then, the TAN of the oil mixture as the refined oil was tested. The deacidification rate was calculated according to the formula below:

$$\text{Deacidification rate} = (1 - \frac{\text{TAN of esterificated oil}}{\text{TAN of raw oil}}) \times 100 \tag{1}$$

To investigate the catalyst lifetime, the catalyst and mixture were separated after the esterification reaction under the optimal reaction conditions, and then the catalyst was reused in the next experiment according to the same optimal reaction conditions, without any further treatment of the catalyst.

**4. Conclusions**

In this work, a series of $Al_2O_3$-supported ZnO with and without $Fe_2O_3$-doping catalysts were synthesized under the optimal preparation conditions: calcination time of 45 min, calcination temperature of 400 °C, impregnation time of 12 h, with zinc oxide loading of 15%, and iron oxide loading of 0.5% using incipient wetness impregnation. Catalytic esterification is a very effective process in the deacidification of high-TAN vacuum gas oil. Significant reduction of the raw VGO TAN from 2.51 to 0.06 mg-KOH/g-oil with a TAN reduction efficiency of 97.6% was achieved by using the catalyst $Fe_2O_3$-ZnO/$Al_2O_3$. The major mechanism of deacidification was esterification of the carboxylic acid group with glycol.

**Author Contributions:** Data curation, X.L. and P.X.; Investigation, X.F., S.J., X.D. and C.H.; Project administration, B.H.; Supervision, C.C.

**Funding:** This research was funded by the Research Program of Chongqing Municipal Education Commission (KJ1713342), National Natural Science Foundation of China (21801031) and the Chongqing Undergraduate Science and Technology Innovation Training Program (201811551020).

**Acknowledgments:** The authors thank Jinyuan Zhang and Shuang Du for their help in the experiments.

**Conflicts of Interest:** The authors declare no conflict of interest.

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
