# Peer review of "Catalytic Deacidification of Vacuum Gas Oil by ZnO/Al2O3 and Its Modification with Fe2O3"

_catalysts, doi:10.3390/catal9060499_

Round 1
Reviewer 1 Report
General comments:
This manuscript “Catalytic deacidification of vacuum gas oil by ZnO/Al2O3 and its modification with Fe2O3” is centered on the development of the general concept of catalytic esterification for deacidification of the vacuum gas oil(VGO).The results reported are of interest to the general community.
The work contains a good structural characterization of all prepared catalysts. This work is very interesting, and the chemistry is O.K.
In summary, after including the following information, the manuscript could be recommended for publication in Catalysts Journal.
Comment 1:
The life of the catalyst was evaluated by performing different uses of the same catalyst. Were leaching tests done? In this work were considered metals as zinc and iron and also very small amounts in solution could give homogeneous catalysis. Please check this aspect.
Comment 2:
In this work were chemical analyzes done to check the quantity of metals present on the catalysts? The values reported are analyzed or nominal values? Please check this aspect.
Comment 3:
The quality of the written English could be improved. Please consider using a native English speaker or an English-language editing service to assist with this.
Author Response
Comment 1:
The life of the catalyst was evaluated by performing different uses of the same catalyst. Were leaching tests done? In this work were considered metals as zinc and iron and also very small amounts in solution could give homogeneous catalysis. Please check this aspect.
√ Thanks very much for your comment. We did not consider the leaching effect of the Zinc and Iron. The catalysts exhibited excellent catalytic activity after used for 6 times without any further treatment which indicated the highly stable character of the catalysts. So the leaching concentration of the active component could be neglected. We will try to detect the ions in the mixture in the further research according to the reviewer’s comment.
Comment 2:
In this work were chemical analyzes done to check the quantity of metals present on the catalysts? The values reported are analyzed or nominal values? Please check this aspect.
√ Thanks very much for your comment. The loading amount of the metals was calculated by precursor and the related metal oxide when preparing the catalyst.
Comment 3:
The quality of the written English could be improved. Please consider using a native English speaker or an English-language editing service to assist with this.
√ Thanks very much for your comment. Thank you so much for your advice and we have improved the written English very carefully which we hope the revised version can meet the standard of the journal.

Reviewer 2 Report
In the work "Catalytic deacidification of vacuum gas oil by ZnO/Al2O3 and its modification with Fe2O3" the deacidification of a vacuum gas oil by several Al2O3/ZnO prepared under different conditions and, subsequently, modified with Fe2O3 as catalysts is addressed.
In the catalyst preparation section, the calcination program(s) should be mentioned.
TAN reduction equation looks like the conversion equation, amazing!!
For the catalyst lifetime test, did the catalyst undergo any treatment (washing, drying, etc.? If yes, indicate which one. If not, mention that the catalyst was reused without further treatment.
In section 3.1.1. it is mentioned that "The crystallinity increased with the calcination time, because it needs enough time for the decomposition of Zn(NO3)2 into ZnO, as well as the growth of the crystal." Why the complete calcination of the material was not allowed? Please, indicate calcination temperature.
Authors state that 400ºC is the best calcination temperature due to a higher crystallinity of the sample based on XRD diagrams. If the XRD plots for the remaining temperatures follow a similar trend. What is causing the different behaviour in the sample calcined at 400 ºC.
Which one was ZnO phase? Which one was Fe2O3 phase?
In section 3.1.2. Which support has been selected for the evaluation of the textural properties? The one calcine at 400 ºC.
On figure 2 it is not possible to see clearly the different isotherm plots. For instance, authors could choose the most representative materials (3 of them) and represent separately their respective isotherm plots.
Figure 3 seems to have been moved when editing, letters do not correspond with images. Please, amend it.
In section 3.2.1, the influence of the calcination temperature is evaluated, it is mentioned the the decrease in deacidification rate is due to the ZnO crystal growth. This statement must be supported by experimental data: TEM or Scherrer equation to evaluate the particle size.
Regarding with the calcination temperature effect, how it has been assessed the decomposition of Zn salt precursor?
Finally, for a reaction where the material acidity plays a key role it is MANDATORY to make experiments to evaluated: amount, strength and type of the acid sites in the materials.
Thus, the recommendation is to be reconsidered after major revisions.
Author Response
1. In the work "Catalytic deacidification of vacuum gas oil by ZnO/Al2O3 and its modification with Fe2O3" the deacidification of a vacuum gas oil by several Al2O3/ZnO prepared under different conditions and, subsequently, modified with Fe2O3 as catalysts is addressed.
In the catalyst preparation section, the calcination program(s) should be mentioned.
√ Thanks very much for your comment. The calcinations program has been added in the revised manuscript according to the comment.
2. TAN reduction equation looks like the conversion equation, amazing!!
√ Thanks very much for your comment. Thanks so much for the comment, and we have modified the equation accordingly.
3. For the catalyst lifetime test, did the catalyst undergo any treatment (washing, drying, etc.? If yes, indicate which one. If not, mention that the catalyst was reused without further treatment.
√ Thanks very much for your comment. The catalyst didn’t undergo any treatment. It was used for the next run just after the reaction and separated. In the revised article, we have added the details.
4. In section 3.1.1. it is mentioned that "The crystallinity increased with the calcination time, because it needs enough time for the decomposition of Zn(NO3)2 into ZnO, as well as the growth of the crystal." Why the complete calcination of the material was not allowed? Please, indicate calcination temperature.
√ Thanks very much for your comment. In fact, the complete calcinations of the nitrate is necessary, but too long time calcinations under the high temperature would lead to the sintering of the catalysts and result in the decrease of surface area and acid sites et al.
5. Authors state that 400ºC is the best calcination temperature due to a higher crystallinity of the sample based on XRD diagrams. If the XRD plots for the remaining temperatures follow a similar trend. What is causing the different behaviour in the sample calcined at 400 ºC.
√ Thanks very much for your comment. Calcination temperature is an important factor for the crystallinity of the sample according to the XRD results. In addition, calcination time is another factor which has significant effect on the crystallinity, surface area and acid sites of the catalysts according to our previous work.
6. Which one was ZnO phase? Which one was Fe2O3 phase?
√ Thanks very much for your comment. We have marked the two phases in the revised manuscript.
7. In section 3.1.2. Which support has been selected for the evaluation of the textural properties? The one calcine at 400 ºC.
√ Thanks very much for your comment. Yes, the Al2O3 support calcine at 400 ºC was chosen overall our research.
8. On figure 2 it is not possible to see clearly the different isotherm plots. For instance, authors could choose the most representative materials (3 of them) and represent separately their respective isotherm plots.
√ Thanks very much for your comment. We have modified Figure 2 according to your suggestion.
9. Figure 3 seems to have been moved when editing, letters do not correspond with images. Please, amend it.
√ Thanks very much for your comment. We have corrected it.
10. In section 3.2.1, the influence of the calcination temperature is evaluated, it is mentioned the decrease in deacidification rate is due to the ZnO crystal growth. This statement must be supported by experimental data: TEM or Scherrer equation to evaluate the particle size.
√ Thanks very much for your comment. We have adjusted the description in original section 3.2.1(2.2.1 now). It was clearly shown in Figure 3 (b) and Figure3 (d),ZnO crystals were quite different. In fact,TEM results would be very good ,but sorry for the short revision time, we will do that evaluation in our further work. We do appreciate your constructive advices.
11. Regarding with the calcination temperature effect, how it has been assessed the decomposition of Zn salt precursor?
√ Thanks very much for your comment. The decomposition of the Zn salt precursor was evaluated by TGA and the results (the figure is in the attachment). Obviously, the decomposition temperature is about 350oC, and a little higher temperature 400oC is benefit for the completely decomposition.
12. Finally, for a reaction where the material acidity plays a key role it is MANDATORY to make experiments to evaluated: amount, strength and type of the acid sites in the materials.
√ Thanks very much for your comment. We figured out the NH3-TPD profiles of the catalysts, and the results were also listed in Figure 4 and Table 2 in the revised article. Results show that there are both weak and strong acid sites on the surface of the catalysts.

Reviewer 3 Report
Authors ought to modify respectively the below points:
Abstract:
Better use of English in the three (3) first lines.
Introduction:
“Catalytic hydrogenation” there is no need for caps.
3.1.2.:
“It was because that the ZnO particles covered part of the small pores, and leaded to reduction of pore surface area and pore volume.” Better use of English.
“This is because more Fe2O3 benefit for forming bigger crystal, which has less BET surface areas and pore volumes than the small particles.” Syntax error.
3.2.2.:
Change the type of the graph in Figure 5. It would be better illustrative to use lines than bars for rates.
3.2.4.:
“Fe2O3 addition could help to improve the esterification which was because the surface acid site on Fe2O3 was the weak acid, and the amount of surface acid on the Fe2O3 was more than that of ZnO.” The meaning of this passage is not clear enough.
3.4:
The authors ought to replace the figure 11 with a table for better appearance of the results, due to the small deviation among the values.
Conclusions:
Repair punctuation errors.
Author Response
Comments and Suggestions for Authors
Authors ought to modify respectively the below points:
1. Abstract:
Better use of English in the three (3) first lines.
√ Thanks very much for your comment. The first three lines were refined according to the comment. In addition, the overall abstract has been rewritten.
2. Introduction:
“Catalytic hydrogenation” there is no need for caps.
√ Thanks very much for your comment. We have corrected it
3. “It was because that the ZnO particles covered part of the small pores, and leaded to reduction of pore surface area and pore volume.” Better use of English.
√ Thanks very much for your comment. We have already corrected it
4. “This is because more Fe2O3 benefit for forming bigger crystal, which has less BET surface areas and pore volumes than the small particles.” Syntax error.
√ Thanks very much for your comment. We have already corrected it
5. Change the type of the graph in Figure 5. It would be better illustrative to use lines than bars for rates.
√ Thanks very much for your comment. Thank you very much for the advice, we tried to change the graph into lines but it seems that the figure looks better.
6. “Fe2O3 addition could help to improve the esterification which was because the surface acid site on Fe2O3 was the weak acid, and the amount of surface acid on the Fe2O3 was more than that of ZnO.” The meaning of this passage is not clear enough.
√ Thanks very much for your comment. We have adjusted the description of that passage.
7. The authors ought to replace the figure 11 with a table for better appearance of the results, due to the small deviation among the values.
√ Thanks very much for your comment. We read lots of the related literatures and found that most of them use bars other than lines. We also tried the lines but it seems that figure looks better than a single table, so we finally selected the figure.
8. Conclusions:
Repair punctuation errors.
√ Thanks very much for your comment. we have already revised them.
